# FLOOD AND ECHO: ALGORITHMIC ALIGNMENT OF GNNS WITH DISTRIBUTED COMPUTING

## ABSTRACT

Graph Neural Networks are a natural fit for learning algorithms. They can directly represent tasks through an abstract but versatile graph structure and handle inputs of different sizes. This opens up the possibility for scaling and extrapolation to larger graphs, one of the most important advantages of an algorithm. However, this raises two core questions i) How can we enable nodes to gather the required information in a given graph (*information exchange*), even if is far away and ii) How can we design an execution framework which enables this information exchange for extrapolation to larger graph sizes (*algorithmic alignment for extrapolation*). We propose a new execution framework that is inspired by the design principles of distributed algorithms: FLOOD AND ECHO Net. It propagates messages through the entire graph in a wave like activation pattern, which naturally generalizes to larger instances. Through its sparse but parallel activations it is provably more efficient in terms of message complexity. We study the proposed model and provide both empirical evidence and theoretical insights in terms of its expressiveness, efficiency, information exchange and ability to extrapolate.

## 1 INTRODUCTION

We study the problem of algorithm learning using Graph Neural Networks. The concept of an algorithm is best understood as a sequence of instructions which can be applied to compute a desired output given the respective input. Algorithms have the advantage, that they work correctly for their entire domain. If we want to multiply two numbers, we can easily illustrate and explain the multiplication algorithm using small numbers. However, the same procedure generalizes, i.e. the algorithm can be used to extrapolate and multiply together much larger numbers using the same algorithmic steps. Algorithm learning aims to grasp these underlying algorithmic principles and incorporate them into machine learning architectures. Therefore, the ability to process different input sizes and extrapolate are at the core of our study.

Graphs and as an extension GNNs naturally present themselves to study algorithms as many algorithmic problems can be represented as graphs. Moreover, they can inherently capture instances of different sizes, which allows us to study extrapolation. GNNs follow the message passing paradigm, which closely corresponds to computation models studied in distributed computing. There it is known, that to derive correct solutions, information has to be exchanged between nodes, often beyond a local neighborhood horizon. This raises two interesting questions regarding the *information exchange* and *algorithmic alignment* for the design of GNNs. How can nodes gather the necessary information in a given graph, even if it is far away? Moreover, is it possible to design an execution framework that is structurally well aligned to facilitate this exchange and enable extrapolation to larger graphs?

We propose a new execution framework, the FLOOD AND ECHO Net. Although, it is still based on message passing, the strategy by which messages are exchanged is very distinct. Not all neighbors exchange messages in every round with all their neighbors. Instead, the execution aligns itself with an algorithm design pattern commonly used in distributed algorithms called *flooding and echo*. There is one node which acts as a starting point, called the root, which initiates a phase consisting of a flooding and an echo part respectively. First, messages are propagated away from the root towards the rest of the graph. In this flooding part, also known as broadcast, nodes only send messages to nodes that are farther away from the root. Once all nodes have received a message, the propagation flow reverses. Now nodes only send messages towards the root, starting with the nodes that are

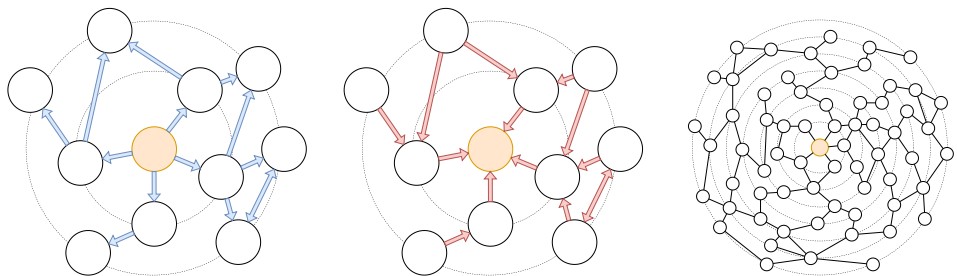

Figure 1: The Flood and Echo Net propagates messages in a wave like pattern throughout the entire graph. Starting from a center node (orange), it floods the messages outwards (blue), then the flow reverses and is echoed back (red). Moreover, the mechanism generalizes naturally to larger graph sizes.

farthest away. This echo part is sometimes also referred to as convergecast. An illustration of this is shown in Figure 1. Note that at every single execution step only a subset of nodes, which are located at the same distance, is activated to send messages either to or away from the root. This activation pattern is at the center of the Flood and Echo Net and can be thought of as a wave pattern which equally extends outwards in all directions from a single point and then reverses back to it.

The Flood and Echo Net is designed to be an overarching execution framework rather than a specific instance of a GNN. It only dictates the information flow of messages, but not how the messages should be aggregated or incorporated into state updates. Therefore, in principle, it can be used in combination with any existing message based graph convolution.

The main advantage of the Flood and Echo Net compared to the standard execution paradigm is that it can more easily facilitate information exchange throughout the entire graph using relatively few messages. The wave like activation of the nodes throughout the graph can be viewed as a special form of asynchronous computation where only a subset of nodes is active at a given time. Messages and node updates are then computed sparsely throughout the graph and happen upon event triggers when receiving a message from a neighbor. This shift towards sparse but simultaneously parallel computation at any given moment is hypothesised to benefit the neural execution on graphs as they align with the execution of parallel algorithms (Engelmayer et al., 2023) or asynchronous neural computation models (Dudzik et al., 2023; Faber & Wattenhofer, 2023).

Standard message passing networks only exchange information with their immediate one hop neighborhood in each round. If only a fixed number of rounds is executed, the information exchange is inherently limited to be local. Therefore, to exchange information throughout the whole graph, an architecture which scales the number of rounds according to the specific graph should be employed. However, the total number of messages grows by the product of the diameter of the graph $D$ and the number of edges $m$ as messages go over all edges in each round, leading to a message complexity of $\mathcal{O}(Dm)$.

On the other hand, the execution of the Flood and Echo Net naturally generalizes to different graph sizes while exchanging messages across the entire graph topology. The number of messages required only scales in terms of the number of edges $\mathcal{O}(m)$. This makes the Flood and Echo Net interesting for the application on graphs across various sizes and especially for the study of extrapolation to larger graph sizes.

We outline our main contributions as follows:

- We introduce Flood and Echo Net, a new execution framework aligned with principles of distributed algorithm design. The computations follows a sparse node activation pattern which allows it to facilitate messages more efficiently.

- We analyze the proposed method both in terms of its expressiveness and efficiency.

- We provide an empirical and theoretical evaluation of Flood and Echo Net's ability to exchange information throughout the entire graph topology.

- We conduct a thorough assessment in the setting of algorithm learning and specifically test the ability to extrapolate to larger graph instances.

## 2 RELATED WORK

Originally proposed by Scarselli et al. (2008) Graph Neural Networks have seen a resurgence with applications across multiple domains (Veličković et al., 2017; Kipf & Welling, 2016; Neun et al., 2022). Notably, this line of research has gained theoretical insights through its connection to message passing models from distributed computing (Sato et al., 2019; Loukas, 2020; Papp & Wattenhofer, 2022a). This includes strengthening existing architectures to achieve maximum expressiveness (Xu et al., 2018; Sato et al., 2021) or going beyond traditional models by changing the graph topology (Papp et al., 2021; Alon & Yahav, 2021b). In this context, multiple architectures have been investigated to combat information bottlenecks in the graph (Alon & Yahav, 2021a), i.e. using graph transformers (Rampasek et al., 2022). Note, that our work is in orthogonal to this, as we focus on staying close to message passing on the original graph topology. Moreover, we investigate how specific information can be exchanged throughout the entire graph, which might be challenging even if no bottleneck is present. Similarly, higher order propagation mechanisms (Zhang et al., 2023b; Maron et al., 2020; Zhao et al., 2022) have been proposed to tackle this issue or gain more expressiveness. While some of these approaches also incorporate distance information, this usually comes at the cost of higher-order message passing. Whereas our work emphasizes a simple execution mechanism on the original graph topology. In recent work, even the synchronous message passing among all nodes has been questioned (Martinkus et al., 2023; Faber & Wattenhofer, 2023), giving rise to alternative neural graph execution models.

GNNs have demonstrated strong generalization capabilities for algorithmic tasks, attributed to their structurally aligned computation (Xu et al., 2020). This has lead to investigate into the proper alignment of parts of the architecture (Dudzik & Veličković, 2022; Engelmayer et al., 2023; Dudzik et al., 2023). A central focus has been on how these networks learn to solve algorithms (Veličković et al., 2022; Ibarz et al., 2022; Minder et al., 2023). Moreover, the ability to extrapolate (Xu et al., 2021) and dynamically adjust the computation in order to reason for longer when confronted with more challenging instances remains a key aspect (Schwarzschild et al., 2021; Grötschla et al., 2022; Tang et al., 2020).

## 3 FLOOD AND ECHO NET

We propose the FLOOD AND ECHO NET, a new execution framework which is designed to align with design patterns from distributed algorithms. The fields ob distributed computing and graph neural network are interlinked through their common usage of message passing-based computation. While the fields are not exactly the same, the connection and equivalence between certain model were established (Papp & Wattenhofer, 2022a). Through this connection, theoretical bounds regarding width, number of rounds and approximation ratios could be translated directly to GNNs (Sato et al., 2019; Loukas, 2020). Moreover, it was discovered, that it is beneficial if an architecture aligns well with the underlying learning objective (Xu et al., 2020; Dudzik & Veličković, 2022) both in terms of performance and sample complexity. This begs the question, if we could potentially translate other insights from distributed computing to the field of graph learning.

First, recall the standard execution of a message passing based GNN. Whenever we refer to a MPNN throughout this paper, we will refer to a GNN which operates on the original graph topology and exchanges messages the following way:

$$a_v^t = \text{AGGREGATE}^k(\{x_u^t \mid u \in N(v)\})$$
$$x_v^{t+1} = \text{UPDATE}(x_v^t, a_v^t)$$

Note that all nodes exchange messages with all their neighbors in every round. We challenge this paradigm, by taking inspiration from a design pattern called *flooding and echo*. This is a common building block in distributed algorithms to first broadcast, or flooding, a piece on information throughout the entire graph and then gather back, or echo, information from all nodes. In principle, this pattern could be replicated using a standard MPNN. However, because nodes send and receive messages at every timestep, throughout the majority of the computation process, nodes would have to learn to remain idle and ignore messages until the relevant flooding or echo messages are received, which makes the learning very challenging.

---

**Algorithm 1** Flood and Echo Net

---

  1: start $\leftarrow$ flood_start
  2: $D \leftarrow$ distances$(G, \text{start})$
  3: maxD $\leftarrow$ max$(D)$
  4: $x \leftarrow$ Encoder$(x)$
  5: **for** t = 0 to phases **do**
  6:     **for** d = 0 to maxD **do**
  7:        $x \leftarrow$ FloodConv$^t(d \rightarrow d + 1, x)$
  8:        $x \leftarrow$ FloodCrossConv$^t(d + 1 \rightarrow d + 1, x)$
  9:     **end for**
10:     **for** d = maxD to 1 **do**
11:        $x \leftarrow$ EchoCrossConv$^t(d \rightarrow d, x)$
12:        $x \leftarrow$ EchoConv$^t(d \rightarrow d - 1, x)$
13:     **end for**
14:     $x \leftarrow$ Update(x)
15: **end for**
16: $x \leftarrow$ Decoder$(x)$

---

We intend to align the overarching computation flow directly with this design pattern. In the Flood and Echo Net, the computation begins from an initial starting node $s$, the root. First, in the flooding part, messages propagate outwards, away from the starting node. Then, once all nodes have received a message, the message flow reverses and is echoed back towards the root. Note that only a subset of nodes, which remain at the same distance, are activated at the same time. Therefore, Flood and Echo Net can make use of a very sparse but simultaneously parallel activation pattern which propagates throughout the graph. We outline a formal definition in the Appendix A. For a more intuitive explanation we refer to Algorithm 1 which outlines a single phase of a Flood and Echo in pseudo code. For a visual explanation, we refer to Figure 1.

**Modes of operation**   Our proposed method is an overarching execution framework, which defines the propagation flow of the computation. However, it does not specify how the specific computation should be done, which allows the usage of any existing graph convolutions inside the framework.

Every execution of the Flood and Echo Net begins with a starting node $s$, from which the propagation pattern follows. This allows us to leverage the Flood and Echo Net in multiple modes of operation for both node and graph classification. This flexibility allows endless combinations of how to choose starts, combine embeddings or even interleaves with other computations. In this work, we only focus on a few specific of the possible modes. For all our modes of operation, we compute node embeddings, which can be directly used for node classification tasks. For graph classification, we sum up the final predicted class probabilities of the individual nodes. Moreover, we differentiate between the *single* and *all* starts mode. In *single* start, either a fixed node is defined to be the start or a node is chosen uniformly at random. We also study the variation, where we choose up to $k$ random nodes, which each execute their computation independent from each other, and then take the mean of their last hidden dimension for all nodes in the graph. This mode allows us to choose a specific (or a few) start, and compute embeddings for all nodes in the graph in a single execution. In the *all* start mode, we execute the Flood and Echo Net once for every node. In every run, we only keep the node embedding for the chosen start node. This can be seen as a form of ego graph prediction for each node, although computationally more expensive, it could potentially be used for efficient inference on node prediction tasks when only a subset of nodes has to be computed. Regardless of the chosen mode, the activation pattern for the Flood and Echo Net remains always the same. It spreads from the starting node, only activating a subset of nodes at a time. This allows the mechanism to efficiently propagate information throughout the entire graph, while simultaneously using few messages.

## 4   THEORETICAL ANALYSIS

This section provides insights into the theoretical properties of the Flood and Echo Net. The presented execution framework is still based on message passing on the original graph topology. However, its message propagation strategy distinguishes it from regular message passing GNNs. Specif-

ically, we will analyze it in terms of expressiveness, where we show that it still captures the mechanisms of other MPNNs while simultaneously going beyond the 1-WL limitation. Furthermore, we analyze the mechanism regarding its efficiency and message complexity.

## 4.1 EXPRESSIVENESS

The expressiveness of GNNs is tightly linked to the WL test (Leman & Weisfeiler, 1968). Most standard message passing architectures, which work on the original graph topology without higher order message passing are limited by the expressiveness of 1-WL (Papp & Wattenhofer, 2022b). However, while our execution is on the original graph topology, it goes beyond 1-WL. Through the alignment of the message propagation with the distance to a specific node in the graph, the mechanism can leverage said information to distinguish more nodes. Moreover, the Flood and Echo Net can still capture the execution of other MPNNs and therefore remains maximally expressive in terms of 1-WL GNNs. To get an intuition on this insight, we can think about how the flooding and echo mechanism differs from the perspective of a single node. Usually, all edges send and receive messages in every round, therefore, the edges are identical. In the Flood and Echo Net, we introduce a "direction" of the edges and we can distinguish between edges that go to the root, away from the root and cross edges which go to nodes located at the same distance. This gives us more possibilities to distinguish nodes in our neighborhood, and even globally, as the wave pattern transitions through the whole graph before it reverses. However, we could chose to ignore this additional directionality information of the edges, which would default to standard MPNN. We can express this finding more formally using Theorem 4.1.

**Theorem 4.1.** *Flood and Echo Net is at least as expressive as any MPNN of width $\mathcal{O}(d)$. Furthermore, it also has width $\mathcal{O}(d)$ while exchanging at most as many messages.*

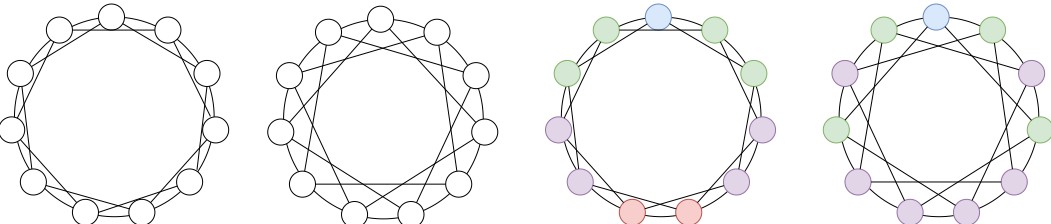

Figure 2: Example of two 4-regular graphs which cannot be distinguished using standard MPNNs as the are 1-WL equivalent. However, no matter which starting point is chosen, the Flood and Echo Net can easily distinguish between them through the derived distance to the starting node.

**Theorem 4.2.** *Flood and Echo Net is strictly more expressive than 1-WL and by extension standard MPNN.*

We empirically validate our findings for the Flood and Echo Net on multiple expressive datasets which go beyond 1-WL. The tasks span both graph and node predictions, which include graphs which have multiple disconnected components. We test two modes on these datasets. One variant performs an execution from a single node (which we refer to as single start) while in the other variant performs an execution once for every node as the starting node(which we refer to as all start). These are directly used for all node prediction tasks, whereas for graph prediction tasks the sum of all node class predictions is used for the final graph prediction, which can be seen as a form of weighted majority. Note that the second variant is more fair for comparison against MPNNs, since for some datasets like Limits-1, Limits-2 and 4-Cycles the graph is not connected. Therefore, the single start mode struggles, as it cannot access all components.

We see that the Flood and Echo all starts manages to almost perfectly solve all tasks. The single start performs worse in the Limits-1 and Limits-2 due to the lack of access to all components. The GIN model, as expected by theory, performs no better than random guessing. The higher scores in the Triangles and LCC datasets are due to the fact that these datasets are imbalanced. For an in depth explanation of the indiviual datasets we refer to Appendix G. Comparing the message complexities of the different methods, a GIN with $L$ layers exchanges $\mathcal{O}(Lm)$ messages while the Flood and

Table 1: As the theory predicts, the GIN model cannot go beyond trivial performance. Whereas both the *single* and *all* execution mode go beyond the limits of 1-WL. Note, that the datasets are imbalanced and can contain multiple components, which can explain the performance of GIN and the drop of the single mode compared to the all execution.

| Model | GIN | | FLOOD AND ECHO *single* | | FLOOD AND ECHO *all* | |
|---|---|---|---|---|---|---|
| | Train | Test | Train | Test | Train | Test |
| Triangles | $0.80 \pm 0.00$ | $0.78 \pm 0.00$ | $0.92 \pm 0.00$ | $0.92 \pm 0.00$ | $1.00 \pm 0.00$ | $1.00 \pm 0.00$ |
| LCC | $0.79 \pm 0.00$ | $0.79 \pm 0.00$ | $0.92 \pm 0.00$ | $0.91 \pm 0.00$ | $1.00 \pm 0.00$ | $1.00 \pm 0.00$ |
| 4-Cycles | $0.49 \pm 0.02$ | $0.50 \pm 0.00$ | $0.95 \pm 0.01$ | $0.95 \pm 0.02$ | $1.00 \pm 0.00$ | $0.96 \pm 0.02$ |
| Limits-1 | $0.50 \pm 0.00$ | $0.50 \pm 0.00$ | $0.70 \pm 0.06$ | $0.80 \pm 0.27$ | $1.00 \pm 0.00$ | $1.00 \pm 0.00$ |
| Limits-2 | $0.50 \pm 0.00$ | $0.50 \pm 0.00$ | $0.79 \pm 0.05$ | $0.90 \pm 0.22$ | $1.00 \pm 0.00$ | $1.00 \pm 0.00$ |
| Skip-Circles | $0.10 \pm 0.00$ | $0.10 \pm 0.00$ | $1.00 \pm 0.00$ | $1.00 \pm 0.00$ | $1.00 \pm 0.00$ | $1.00 \pm 0.00$ |
| Messages | $\mathcal{O}(Lm)$ | | $\mathcal{O}(m)$ | | $\mathcal{O}(nm)$ | |

Echo model either exchanges $\mathcal{O}(m)$ or $\mathcal{O}(nm)$ messages based on whether it executes the single or all starts mode.

## 4.2 MESSAGE COMPLEXITY

We analyze our proposed methods from a theoretical perspective in terms of efficiency. Often, efficiency of message passing is measured as the number of rounds or number of exchanged messages. Recall, that during a single round of a regular MPNNs messages are sent over each edge, meaning $\mathcal{O}(m)$ messages are exchanged every round. However, in a single round of Flood and Echo Net only a subset of nodes is active, as messages are either sent away or towards the starting node. This sparse activation of nodes throughout the graph, allows the mechanism to take full advantage of sending comparatively few messages which can propagate throughout the entire graph. Only after an entire phase of flooding and echoing back, $\mathcal{O}(m)$ messages have been exchanged. Note, that in terms of number of messages, a single phase seems equivalent to a single round of an MPNN. However, In the Flood and Echo model, these messages propagate throughout the entire graph, potentially exchanging information up to $\mathcal{O}(D)$ hops away. In comparison, a single round of an MPNN only exchanges information for 1 hop at a time. We can leverage this discrepancy and formally prove, that using the Flood and Echo execution framework there exists tasks which can be solved much more efficiently. Moreover, recall that by Theorem 4.1, we could simulate the execution of other MPNNs while using at most the same number of messages.

**Lemma 4.3.** *There exist tasks, which Flood Echo can solve using $\mathcal{O}(m)$ messages whereas no MPNN can solve this using less than $\mathcal{O}(nm)$ messages.*

## 5 INFORMATION PROPAGATION

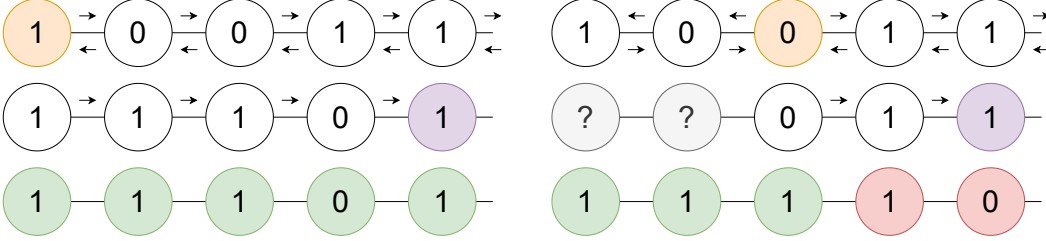

Figure 3: Visualization of the information exchange in the PrefixSum task when choosing different starting nodes for Flood and Echo Net. We can derive theoretical upper bounds for the performance of Flood and Echo Net depending on the number of random start nodes for a single phase. We show that the empirical performance closely follows the theoretical analysis. This confirms the ability of the Flood and Echo Net to distribute the available information throughout the whole graph.

Table 2: Information propagation of the Flood and Echo Net for graphs of size $n$ on the PrefixSum task. As the number of random starting points $s$ increases, the model can distribute the additional information, seen by the increase in accuracy. Moreover, it can do so very effectively as the performance closely follows the theoretical upper bound.

| Model | $n = 10$ | | | | $n = 100$ | | | |
|---|---|---|---|---|---|---|---|---|
| | $s = 1$ | $s = 2$ | $s = 3$ | $s = 5$ | $s = 1$ | $s = 2$ | $s = 3$ | $s = 5$ |
| THEORETICAL UPPER BOUND | 82.00 | 89.80 | 93.52 | 96.91 | 75.75 | 84.07 | 88.23 | 92.39 |
| FLOOD AND ECHO | $81.69 \pm 0.51$ | $88.10 \pm 2.34$ | $89.99 \pm 0.28$ | $93.90 \pm 0.23$ | $75.39 \pm 0.29$ | $83.43 \pm 0.44$ | $87.79 \pm 0.34$ | $91.86 \pm 0.28$ |

In this section, we analyze the ability of the Flood and Echo Net to distribute the available information throughout the whole graph. We use a synthetic algorithmic dataset, the PREFIXSUM task. For this task, we can provably determine what pieces of information must be gathered for each node to make correct predictions. If we choose an appropriate starting point, we could easily send the information and solve the task. However, more interestingly, what happens if we choose a random starting node instead? Can the Flood and Echo model still distribute the relevant information, even if it does not suffice to fully solve the task? We derive theoretical upper bounds for the best performing instance given the information which theoretically could be available during the execution depending on the number of starting nodes. Interestingly, even if the full information is not available, the Flood and Echo Net achieves performance which closely follows the theoretical upper bound. This showcases the ability of out proposed method to distribute all available information throughout the whole graph.

**PrefixSum Task**    For this analysis, we use the PrefixSum dataset, which follows the task introduced by Schwarzschild et al. (2021) and was later adapted for the graph setting (Grötschla et al., 2022). It consists of a path graph, where one end is marked to distinguish left form right. Each node $v$ independently and uniformly at random gets assigned one bit $x_v$, which is either 1 or 0, chosen with probability $\frac{1}{2}$ each. The task is to compute the prefix sum from left to right modulo 2. Therefore, the output $y_v$ of each node $v$ is the sum of the bits of all nodes to the left $y_v \equiv_2 \left( \sum_{i \leq v} x_i \right)$. Note, that in order to correctly predict a node output, it has to take all bits left of it into consideration.

**Lemma 5.1.** *In the PrefixSum task, for every node $v$, the computation of the output $o_v$ must be dependent on all bits of the nodes to its left. If not all bits are considered for the computation, the probability of a correct prediction is bounded by $\Pr[o_v = y_v] \leq \frac{1}{2}$.*

Note that from this lemma it immediately follows that for solving the task correctly you need to exchange information throughout the whole graph. Nodes towards the end of the path must consider almost all nodes of the graph for their computation.

**Corollary 5.2.** *The PrefixSum task requires information of nodes that are $\mathcal{O}(D)$ hops apart and therefore must exchange information throughout the entire graph.*

From Lemma 5.1, we know that nodes can only correctly predict their output, if the information of all nodes left to them is taken into account. Whenever the initial starting point of the Flood and Echo Net is chosen at one of the ends, this information should be available in either the flooding or echo part. However, what happens if we choose one of the nodes in the graph at random to be the starting point? Then, there will always be a right side, whose predictions are dependent on the computation of the left which was not yet exchanged. An example is depicted in Figure 3. The top row indicates the starting node (orange) and illustrates the message exchange in the flooding (top arrows) and echo phase (bottom arrows). The middle row indicates what parts of the graph the purple marked node can know about after a single phase. Note, that on the right hand side it cannot infer the initial features of the two left most nodes. Because on the missing information, the configuration on the right can only correctly predict the nodes up to the initial starting node.

We can formally derive a theoretical upper bound for the expected number of correctly predicted nodes depending on $n$, the number of nodes and $s$, the number of starts. Let us assume $s$ starting nodes are chosen uniformly at random and $s_j$ denote the index of the $j$-th starting nodes. If the beginning is chosen, then all nodes could be classified correctly. Otherwise, nodes can only be correctly inferred up to $t = \max_j s_j$, the starting node farthest to the right. Moreover, the rest of the $n - t$ nodes can only be guessed correctly with probability $\frac{1}{2}$ as the cumulative sum to the left

is missing. We can write down a closed form solution for $X$, the expected number of correctly predicted nodes for a perfect solution as follows:

$$\mathbb{E}[X] = \Pr_j[\min_j s_j = 1]n + (1 - \Pr_j[\min_j s_j = 1]) \sum_{i=2}^{n} \frac{n + \max_j s_j}{2} \Pr_j[\max_j s_j = i]$$

$$= \left(1 - \left(\frac{n-1}{n}\right)^s\right)n + \left(\frac{n-1}{n}\right)^s \sum_{i=2}^{n} \frac{n+i}{2}(\Pr_j[\max_j s_j < i+1] - \Pr_j[\max_j s_j < i])$$

$$= \left(1 - \left(\frac{n-1}{n}\right)^s\right)n + \left(\frac{n-1}{n}\right)^s \sum_{i=2}^{n} \frac{n+i}{2}\left(\left(\frac{i-1}{n-1}\right)^s - \left(\frac{i-2}{n-1}\right)^s\right)$$

In Table 2, we can compare the empirical performance of the Flood and Echo Net with the theoretical upper bound. Moreover, the measured performance closely follows the theoretical upperbound. The experiment clearly shows, that the accuracy of the model strictly increases when more starting nodes are chosen. This indicates, that the model can make use of the additional provided information. Therefore, it can effectively incorporate the information and propagate it in a sensible way throughout the graph.

## 6 EXTRAPOLATION

In this section, we study algorithm learning on three different tasks. Our main objective is to learn a solution, which can imitate the behaviour of a classical algorithm and generalize to new unseen instances. In particular, extrapolation to larger graph sizes beyond the seen training data.

The three tasks we study are the *PrefixSum*, *Distance* and *Path Finding* task, used by Grötschla et al. (2022). In the Distance task, nodes have to infer their distance to a marked node modulo 2. For the *Path Finding* task, nodes in a tree have to predict whether they are part of the path between two given nodes. For a more detailed description of the datasets at hand, we refer to Appendix F. Similarly to the PrefixSum task, the tasks provably require the exchange of information throughout the whole graph in order to solve the tasks correctly.

**Corollary 6.1.** *Both the Distance and Path Finding task require information of nodes that are $\mathcal{O}(D)$ hops apart and therefore must exchange information throughout the entire graph.*

We consider the three main modes of operation of the Flood and Echo Net presented in this work: *all*, where each node acts as a starting node once to predict its own label. Furthermore, we consider the *single* mode, both for one random starting node and for the fixed starting node. We hypothesize, that in many settings, it makes sense to fix a starting node a priori from the given task requirements, i.e. for shortest paths it is natural to consider the source as the starting node. All Flood and Echo models execute two phases, which results in $\mathcal{O}(m)$ messages exchanged per starting node. We also consider GIN, a representative of a maximal expressive MPNN, as a baseline. We fix the number of

Table 3: Extrapolation experiments on all algorithmic datasets, all models were trained with graphs of size 10 and then tested on larger graphs of size 100. We compare the different Flood and Echo models against a regular GIN, which executes $L$ rounds and RecGNN, which adapts the number of rounds. The *random* mode picks a starting node at random, while the *fixed* mode starts at a predefined location. The *all* chooses each node as a start once. We report both the node accuracy with $n()$ and the graph accuracy with $g()$.

| Model | MESSAGES | PREFIXSUM | | | DISTANCE | | | PATH FINDING | | |
|---|---|---|---|---|---|---|---|---|---|---|
| | | n(10) | n(100) | g(100) | n(10) | n(100) | g(100) | n(10) | n(100) | g(100) |
| GIN | $\mathcal{O}(Lm)$ | $0.78 \pm 0.01$ | $0.53 \pm 0.00$ | $0.00 \pm 0.00$ | $0.97 \pm 0.01$ | $0.91 \pm 0.01$ | $0.04 \pm 0.06$ | $0.99 \pm 0.01$ | $0.70 \pm 0.05$ | $0.00 \pm 0.00$ |
| RecGNN | $\mathcal{O}(nm)$ | $1.00 \pm 0.00$ | $0.93 \pm 0.07$ | $0.66 \pm 0.31$ | $1.00 \pm 0.00$ | $0.99 \pm 0.02$ | $0.93 \pm 0.15$ | $1.00 \pm 0.00$ | $0.95 \pm 0.04$ | $0.45 \pm 0.33$ |
| Flood and Echo *all* | $\mathcal{O}(nm)$ | $1.00 \pm 0.00$ | $1.00 \pm 0.01$ | $0.96 \pm 0.07$ | $1.00 \pm 0.00$ | $0.99 \pm 0.03$ | $0.87 \pm 0.25$ | $1.00 \pm 0.00$ | $0.92 \pm 0.05$ | $0.14 \pm 0.22$ |
| Flood and Echo *random* | $\mathcal{O}(m)$ | $1.00 \pm 0.00$ | $1.00 \pm 0.00$ | $0.99 \pm 0.01$ | $1.00 \pm 0.00$ | $0.97 \pm 0.04$ | $0.77 \pm 0.30$ | $1.00 \pm 0.00$ | $0.82 \pm 0.01$ | $0.01 \pm 0.00$ |
| Flood and Echo *fixed* | $\mathcal{O}(m)$ | $1.00 \pm 0.00$ | $1.00 \pm 0.00$ | $1.00 \pm 0.00$ | $1.00 \pm 0.00$ | $1.00 \pm 0.00$ | $0.99 \pm 0.02$ | $1.00 \pm 0.00$ | $1.00 \pm 0.00$ | $1.00 \pm 0.00$ |

Table 4: Extrapolation on the PrefixSum task. All models are trained with graphs of size 10 and then tested on larger graphs. The Flood and Echo models are able to generalize well to graphs 100 times the sizes encounterd during training. We report both the node accuracy with $n()$ and the graph accuracy with $g()$.

| Model | MESSAGES | PREFIXSUM | | | | | |
| --- | --- | --- | --- | --- | --- | --- | --- |
| | | n(10) | g(10) | n(100) | g(100) | n(1000) | g(1000) |
| GIN | $\mathcal{O}(Lm)$ | $0.78 \pm 0.01$ | $0.07 \pm 0.03$ | $0.53 \pm 0.00$ | $0.00 \pm 0.00$ | $0.50 \pm 0.00$ | $0.00 \pm 0.00$ |
| RecGNN | $\mathcal{O}(nm)$ | $1.00 \pm 0.00$ | $1.00 \pm 0.00$ | $0.93 \pm 0.07$ | $0.66 \pm 0.31$ | $0.72 \pm 0.24$ | $0.40 \pm 0.52$ |
| Flood and Echo *random* | $\mathcal{O}(m)$ | $1.00 \pm 0.00$ | $1.00 \pm 0.00$ | $1.00 \pm 0.00$ | $0.99 \pm 0.01$ | $1.00 \pm 0.00$ | $0.89 \pm 0.10$ |
| Flood and Echo *fixed* | $\mathcal{O}(m)$ | $1.00 \pm 0.00$ | $1.00 \pm 0.00$ | $1.00 \pm 0.00$ | $1.00 \pm 0.00$ | $1.00 \pm 0.00$ | $1.00 \pm 0.00$ |

rounds to be constant, in this specific case five rounds are executed as the model began to destabilize for more rounds. Note, that from the insights of Corollary 6.1, it follows that all GNNs which solve the task correctly cannot limit themselves to execute $\mathcal{O}(1)$ rounds of message passing. Therefore, we cannot expect good extrapolation performance from the GIN.

**Corollary 6.2.** *Every MPNN which correctly solves the PrefixSum, Distance or Path Finding for all graph sizes $n$ must execute at least $\mathcal{O}(D)$ rounds and exchange $\mathcal{O}(mD)$ messages.*

Therefore, we consider another baseline, RecGNN, a recurrent architecure by the design of Grötschla et al. (2022). This GNN can dynamically adjust the number of executions depending on the graph size. This, in principle, would allow the model to exchange the required information throughout the graph. However, note that in each step all nodes exchange messages with their neighbors for up to $\mathcal{O}(n)$ rounds. The model has to retain stability across many iterations, whereas our Flood and Echo models can alleviate this potential issue through sparse activations during the computation.

**Lemma 6.3.** *Flood and Echo Net can facilitate the required information for the PrefixSum, Distance and Path Finding task in a single phase, which can be executed using $\mathcal{O}(m)$ messages.*

In Table 3, we can compare the performance of the aforementioned baselines. Note, all models are trained on graphs of size 10. We test how well they have learned the underlying algorithmic pattern of the tasks and evaluate on graphs of size 100. We observe that the baseline with a fixed number of layers already struggles to fit the training data and deteriorates when tested on larger instances. The other models exhibit better generalization, however, especially the node accuracy remains quite high. Unfortunately, this is a bit deceiving. In algorithm evaluation, we expect them to be correct for a complete problem instance. Otherwise, even if only a single node is mispredicted, the algorithm is not correct for the instance. Therefore, we also report the graph accuracy, which measures how many graph instances are predicted correctly, i.e. all nodes within the graph were assigned their correct label. There, we can see that the overall model performance of the baselines drop compared to the Flood and Echo. Moreover, we can test extrapolation to even larger instances as shown in Table 6. Note that even though the node accuracy for many entries is quite high, the graph accuracy deteriorates as the graph sizes increases. The Flood and Echo models seem to be more robust to this process, especially when the starting node is chosen in a fixed manner. This showcases, that the proposed execution model is well aligned for extrapolation. It can generalize across graph sizes and for some settings even perfectly to much larger graphs that are far beyond the training sizes.

## 7 CONCLUSION

We present a new simple execution framework, the Flood and Echo Net whose mechanism is aligned with algorithm design principles from distributed computing. Instead of exchanging messages everywhere at once, the message flow propagates in a wave like activation throughout the entire graph. This distance centered sparse activation of the nodes provably increases both the efficiency and expressiveness of our method. In a controlled environment, we thoroughly analyze its ability to facilitate information throughout the graph both from a theoretical and empirical perspective. Moreover, we find that our proposed execution strategy naturally generalizes across graph sizes. Specifically, the Flood and Echo Net seems to algorithmically align well for extrapolation. This enables us to do algorithm learning on small instances during training and then test on graphs that are up to 100 times larger than during training.

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

## A  FLOOD AND ECHO DEFINITION

Let $r$ be the root of the computation phase and let $d(v)$ denote the shortest path distance from $v$ to $r$. Then the update rule for one phase of the Flood and Echo Net looks as follows:

$$f_v = \text{AGGREGATE}_{Flood}(\{x_u^t \mid d(u) = d(v) - 1, v \in N(v)\})$$
$$x_v^{t+1} = \text{UPDATE}_{Flood}(x_v^t, f_v^t)$$
$$fc_v = \text{AGGREGATE}_{FloodCross}(\{x_u^{t+1} \mid d(u) = d(v), v \in N(v)\})$$
$$x_v^{t+2} = \text{UPDATE}_{FloodCross}(x_v^{t+1}, fc_v^t)$$
$$ec_v = \text{AGGREGATE}_{EchoCross}(\{x_u^{t+2} \mid d(u) = d(v) + 1, v \in N(v)\})$$
$$x_v^{t+3} = \text{UPDATE}_{EchoCross}(x_v^{t+2}, ec_v^t)$$
$$e_v = \text{AGGREGATE}_{Echo}(\{x_u^{t+3} \mid d(u) = d(v), v \in N(v)\})$$
$$x_v^{t+4} = \text{UPDATE}_{Echo}(x_v^{t+3}, e_v)$$
$$x_v^{t+5} = \text{UPDATE}(x_v^{t+4})$$

Note that the node activations are done in a sparse way, therefore, for all updates that take an empty neighborhood set as the second argument no update is performed and the state is maintained. Furthermore, in practise we did not find a significant difference in performing the last update setp, which is why in the implementation we do not include it.

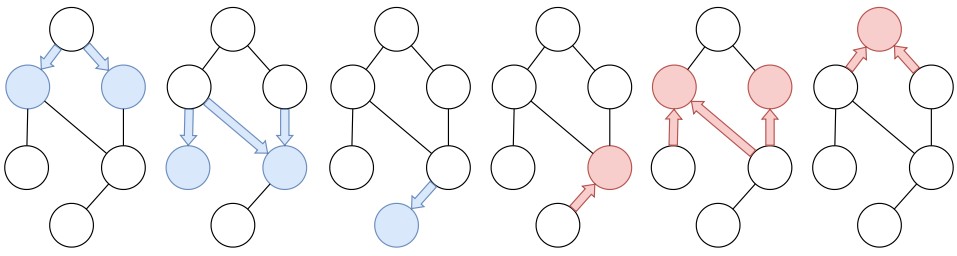

Figure 4: Illustration of the activation pattern of a single phase of a Flood and Echo Net. At every update step only a subset of nodes are active and change their state. The starting node is the top node of the graph and the blue/red arrows depict the information flow in the flood/echo part respectively.

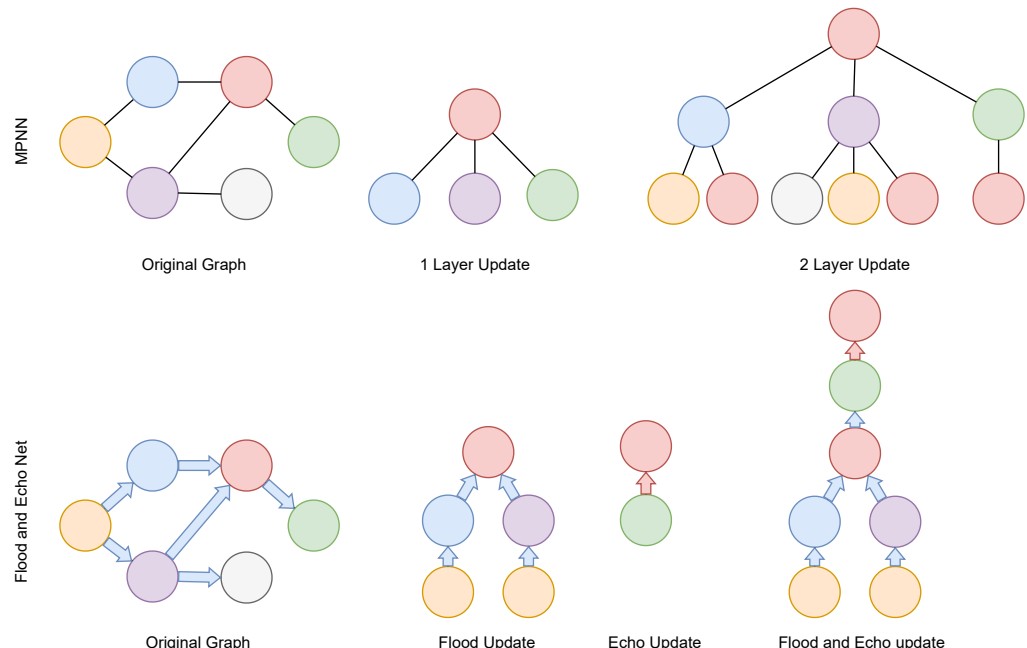

Figure 5: Visualization of the computation executed on the same graph for a regular MPNN and a Flood and Echo Net from the perspective of the red node. The top row shows the computation for regular MPNN both for 1 and 2 layers of message passing. Note that executing l layers takes into account the l-Hop neighborhood. On the bottom rows, the computation from the perspective of the red node in a Flood and Echo net is shown. Note that the starting node is the orange node. The two middle figures illustrate the respective updates in the flood part and the echo part respectively. Furthermore, the figure on the right shows the computation for an entire phase.

## B  EXTENDED RELATED WORK

A variety of GNNs which do not follow the 1 hop neighborhood aggregation scheme have been unified under the view of so called Subgraph GNNs. The work of Zhang et al. (2023a) analyses these models in terms of their expressiveness and gives the following general definition:

**Definition B.1.** *A general subgraph GNN layer has the form*

$$h_G^{(l+1)}(u,v) = \sigma^{(l+1)}(\mathsf{op}_1(u,v,G,h_G^{(l)}), \cdots, \mathsf{op}_r(u,v,G,h_G^{(l)})),$$

*where $\sigma^{(l+1)}$ is an arbitrary (parameterized) continuous function, and each atomic operation $\mathsf{op}_i(u,v,G,h)$ can take any of the following expressions:*

- *Single-point: $h(u,v)$, $h(v,u)$, $h(u,u)$, or $h(v,v)$;*

- *Global: $\sum_{w \in \mathcal{V}_G} h(u,w)$ or $\sum_{w \in \mathcal{V}_G} h(w,v)$;*

- *Local: $\sum_{w \in \mathcal{N}_{G^u}(v)} h(u,w)$ or $\sum_{w \in \mathcal{N}_{G^v}(u)} h(w,v)$.*

*We assume that $h(u,v)$ is always present in some $\mathsf{op}_i$.*

This allows to capture a more general class of Graph Neural Networks, i.e. the work of (Zhang et al., 2023b) which can incorporate distance information into the aggregation mechanism this way. Note, that the proposed mechanism of the Flood and Echo Net differs to that of this particular notion of subgraph GNNs. At each update step, only a subset of nodes is active (illustrated in Figure 4). This allows nodes to take into account nodes that are activated earlier, which is not directly comparable to subgraph GNNs where the node updates still happen simultaneously for the nodes in question.

Another important issue that GNNs often struggle with is the so called phenomena of oversquashing (Alon & Yahav, 2021a). In simple terms, if too much information has to be propagated through

the graph using few edges, a bottleneck occurs which squashes the relevant information together, leading to information loss and subsequent problems for learning. Recent work of (Giovanni et al., 2023) theoretically analyses the reasons leading to the oversquashing phenomena and identify the width and depth of the network but also the graph topology as key contributors. Note that the proposed Flood and Echo Net is not designed to tackle the problem of oversquashing. Rather, it tries to facilitate information throughout the graph, assuming that there is no inherent (topological) bottleneck. It only affects the aforementioned depth aspect of the network. However, as outlined by (Giovanni et al., 2023) the depth is likely to have a marginal affect compared to the graph topology.

## C   PROOFS

*Proof of Theorem 4.1.* It has been shown by the work of Xu et al. (2018) that the *Graph Isomorphism Network* (GIN) achieves maximum expressiveness amongst MPNN. In the following, we will show that a Flood and Echo Net can simulate the execution of a GIN, therefore matching it in its expressive power. Let $G_I$ be a GIN using a node state vector $h_v^k$ of dimension $d_i$.

$$h_v^{(k)} = \text{MLP}^{(k)}((1+\epsilon)h_v^{(k-1)} + \sum_{u \in \mathcal{N}(v)} h_u^{(k-1)})$$

Let $G_F$ be a Flood and Echo Net using node state vector $q_v^{(k)}$ of dimension $d_f = 2 \cdot d_i$. We partition the vector $q_v^{(k)} = o_v^{(k)} \mathbin{\|} n_v^{(k)}$ into two vectors of dimension $d_i$. Initially, we assume that the encoder gives us $o_v^{(0)} = h_v^{(0)}$ and $n_v = 0^{d_i}$ the zero vector. We now define the updates of flood, floodcross, echo and echocross in a special way, that after the flood and echo part $o_v^{(k)}$ is equal to $h_v^{(k)}$ and $n_v^{(k)}$ is equal to $\sum_{u \in \mathcal{N}(v)} h_u^{(k-1)}$. If this is ensured, the final update in a flood and echo phase can update $q_v^{(k)} = \text{MLP}^{(k)}((1+\epsilon)o_v^{(k-1)} + n_v^{(k-1)}) \mathbin{\|} 0^{d_i}$, which exactly mimics the GIN update. It is easy to verify that if we set the echo and flood updates to add the full sum of the $o_v^{(k)}$ part of the incoming messages (and similarly half of the sum of the incoming messages during the cross updates) to $n_v^{(k-1)}$ the desired property is fulfilled. Moreover, there are at most four messages exchanged over each edge of the graph. Specifically, four for cross edges and two for all other edges. Therefore, a total of $\mathcal{O}(m)$ messages are exchanged, which is asymptotically the same number of messages GIN exchanges in a single update step. This enables a single phase of the Flood and Echo Net to mimic the execution of a single GIN round. Repeating this process the whole GIN computation can be simulated by the Flood and Echo Net.

Therefore, given a GIN network $G_I$ of width $d_i$ we can construct a Flood and Echo Net $G_F$ of width $\mathcal{O}(d)$ that can simulate one round of $G_I$ in a single flood and echo phase using $\mathcal{O}(m)$ messages.

$\square$

*Proof of Theorem 4.2.* To show that the Flood and Echo Net goes beyond 1-WL, it suffices to find two different graphs which are equivalent under the 1-WL test, but can be distinguished by a Flood and Echo Net. Observe, that a Flood and Echo Net can calculate for each node its distance, in number of hops, to the root. See the graphs illustrated in Figure 2 for a comparison. On the left is a cycle with 11 nodes which has additional connections to the nodes which are at distance two away. Similarly, the graph on the right has additional connections at a distance of three. Both graphs are four regular and can therefore not be distinguished using the 1-WL test. However, no matter where the starting node for Flood and Echo is placed, it can distinguish that there are nodes which have distance four to the starting root in one graph, which is not the case in the other graph. Therefore, Flood and Echo Net can distinguish the two graphs and is more expressive than the 1-WL test.

$\square$

*Proof of Lemma 4.3.* Consider either one of the PrefixSum, Distance or Path Finding task presented in Appendix F. All of them require information which is $\mathcal{O}(D)$ apart and must be exchanged. It follows, that all MPNNs must execute at least $\mathcal{O}(D)$ rounds of message passing to facilitate this information. Moreover, in these graphs, the graph diameter can be $\mathcal{O}(n)$. As in each round there are $\mathcal{O}(m)$ messages exchanged, MPNNs must use at least $\mathcal{O}(nm)$ messages to solve these tasks.

Furthermore, from Lemma 6.3 it follows that Flood and Echo Net can solve the task in a single phase using $\mathcal{O}(m)$ messages. $\quad\square$

*Proof of Lemma 5.1.* Assume for the sake of contradiction, that not all bits of the nodes to the left have to be taken into consideration for the computation. Therefore, there exists at least one bit at a node $u$ which is not considered for the computation of $o_v$. We know that all bits $x$ are drawn uniformly at random and are independent of each other. Furthermore, we can rewrite the groundtruth $y_v \equiv_2 \sum_{i \leq v} x_i \equiv_2 x_u + \sum_{i \leq v, i \neq u} x_i \equiv_2 x_u + s$ as the sum of $x_u$ and the rest of the nodes. From there it follows, that the groundtruth is dependent on $x_u$, even if all other bits are known $\Pr[y_v = 0 \mid s] = \Pr[s = x_u] = \frac{1}{2}$. On the other hand, we know that $o_v$ must be completely determined by the information of the nodes that make up $s$ and cannot change depending on $x_u$. Therefore, $\Pr[o_v = y_v \mid o_v \text{ does not consider } x_u] \leq \frac{1}{2}$. $\quad\square$

*Proof of Corollary 5.2.* According to Lemma 5.1, for each node $v$ to derive to the correct prediction, all $x_u$ for nodes $u$ that are left of $v$ have to be considered. Therefore, look at the node $r$ on the very right end of the path graph. It has to take the bits of all other nodes into consideration, however, the leftmost bit at node $l$ is $n - 1$ hops away, which is also the diameter of the graph. Therefore, in order to solve the PrefixSum task, information has to be exchanged throughout the entire graph by propagating it for at least $\mathcal{O}(D)$ hops. $\quad\square$

*Proof of Corollary 6.1.* Assume for the sake of contradiction that this is not the case and only information has to be exchanged which is $d' = o(D)$ hops away to solve the task. Therefore, as both tasks are node prediction tasks, the output of each node is defined by its $d'$-hop neighborhood. For both tasks we construct a star like graph $G$ which consists of a center node $c$ and $k$ paths of length $\frac{n}{k}$ which are connected to $c$ for a constant $k$. For the Path Finding task, let the center $c$ be one marked node and the end of path $j$ be the other marked node. Consider the nodes $x_i$, $i = 1, 2, ..., k$ which lie on the $i$-th path at distance $\frac{n}{2k}$ from $c$. Note that all $x_i$ are $\frac{n}{2k}$ away from both their end of the path and $c$ the root. Moreover, the diameter of the graph is $\frac{2n}{k}$. This means that neither the end of the $i$-th path nor the center $c$ will ever be part of the $d'$hop neighborhood. Therefore, if we can only consider the $d'$-hop neighborhood for each $x_i$, they are all the same and as a consequence will predict the same solution. However, $x_j$ lies on the path between the marked nodes while the other $x_i$'s do not. So they should have different solutions, a contradiction. A similar argument holds for the Distance task. Again let $c$ be the marked node in the graph and $x_i$ for $i = 1, 2, ..., k$ be the nodes which lie on the $i$-th path at distance $\frac{n}{2k}$ for even $i$ and $\frac{n}{2k} + 1$ for odd $i$. Again, note that the $d'$-hop neighborhood of all $x_i$ is identical and therefore must compute the same solution. However, the solution of even $x_i$ should be different from the odd $x_i$, a contradiction. $\quad\square$

*Proof of Corollary 6.2.* From Corollary 6.1 and 5.2 it directly follows that information must be exchanged for at least $\mathcal{O}(D)$ hops to infer a correct solution. As MPNNs only exchange information one hop and exchange $\mathcal{O}(m)$ messages per round, the claim follows immediately. $\quad\square$

*Proof of Lemma 6.3.* We will prove that in all three mentioned tasks, there exists a configuration for a Flood and Echo phase, which can propagate the necessary information throughout the graph in a single phase. Let the starting point $s$ correspond to the marked node in the graph, or in the case of the Path Finding any of the two suffices. First, we consider the PrefixSum task. Note that in the flooding phase, information is propagated from the start, which is the left end, towards the right. Therefore, in principle, each bit can be propagated to the right, which suffices to solve the task according to 5.1. For the Distance task, it is necessary that the length of the shortest path between the root and each node can be inferred. Note that this is exactly the path which is taken by the flooding messages, therefore, this should be sufficient to solve the task. Similarly, for the Path Finding task one phase is sufficient. Note that starting from the leaves of the graph during the echo phase, nodes can decide that they are not part of the path between the two marked nodes (as only marked leaves can be part of the path). However, when such a message is received at one of the marked nodes, they can ignore it and tell their predecessor that they are on the path. This is correct as one of the marked ends is at the start of our computation and this echo message travels from the other marked end on the to be marked path back towards the root. This shows, that for each of the above mentioned tasks there exists a Flood and Echo Net configuration which solves the task in a single phase, which exchanges $\mathcal{O}(m)$ messages. $\quad\square$

# D  MODEL ARCHITECTURE AND TRAINING

In our experiments we use a GRUMLP convolution for all Flood and Echo models and the RecGNN which is defined in equation 1. It concatenates both endpoints of an edge for its message and passes it inot a GRU cell (Cho et al., 2014). All models use a hidden node state of 32. We use a multilayer perceptron with hidden dimension 4 times the input dimension and map back to the hidden node state. We use LayerNorm introduced by (Ba et al., 2016). For the expressiveness tasks we perform one phase of Flood and Echo message passing to compute our node embeddings while for the algorithmic tasks we perform two phases of Flood and Echo message passing.

$$x_v^{t+1} = \text{GRUCell}\left(x_v^t, \sum_{u \in N(v)} \phi(x_v^t || x_u^t)\right) \tag{1}$$

In all our experiments we train our model using the ADAM optimizer Kingma & Ba (2015) with a learning rate of $4 \cdot 10^{-4}$ and batch size of 32 for 200 epochs. We also use a learning rate scheduler where we decay the learning rate with a patience of 3 epochs and perform early stopping if the validation loss does not decrease for more than 25 epochs. All reported values are reported over the mean of 5 runs.

The model is implemented in pytorch lightning using the pyg library and the code will be made public upon publication.

# E  EXTRAPOLATION

Table 5: Extrapolation on the Distance task. All models are trained with graphs of size 10 and then tested on larger graphs. The Flood and Echo models are able to generalize well to graphs 100 times the sizes encounterd during training. We report both the node accuracy with $n()$ and the graph accuracy with $g()$.

| Model | MESSAGES | DISTANCE | | | | | |
|---|---|---|---|---|---|---|---|
| | | n(10) | g(10) | n(100) | g(100) | n(1000) | g(1000) |
| GIN | $\mathcal{O}(Lm)$ | $0.99 \pm 0.01$ | $0.92 \pm 0.06$ | $0.70 \pm 0.05$ | $0.00 \pm 0.00$ | $0.53 \pm 0.01$ | $0.00 \pm 0.00$ |
| RecGNN | $\mathcal{O}(nm)$ | $1.00 \pm 0.00$ | $1.00 \pm 0.00$ | $0.95 \pm 0.04$ | $0.45 \pm 0.33$ | $0.78 \pm 0.13$ | $0.00 \pm 0.00$ |
| Flood and Echo *random* | $\mathcal{O}(m)$ | $1.00 \pm 0.00$ | $1.00 \pm 0.00$ | $0.82 \pm 0.01$ | $0.01 \pm 0.00$ | $0.58 \pm 0.01$ | $0.00 \pm 0.00$ |
| Flood and Echo *fixed* | $\mathcal{O}(m)$ | $1.00 \pm 0.00$ | $1.00 \pm 0.00$ | $1.00 \pm 0.00$ | $1.00 \pm 0.00$ | $1.00 \pm 0.00$ | $1.00 \pm 0.00$ |

Table 6: Extrapolation on the Path-Finding task. All models are trained with graphs of size 10 and then tested on larger graphs. The Flood and Echo models are able to generalize well to graphs 100 times the sizes encounterd during training. We report both the node accuracy with $n()$ and the graph accuracy with $g()$.

| Model | MESSAGES | PATH-FINDING | | | | | |
|---|---|---|---|---|---|---|---|
| | | n(10) | g(10) | n(100) | g(100) | n(1000) | g(1000) |
| GIN | $\mathcal{O}(Lm)$ | $0.97 \pm 0.01$ | $0.77 \pm 0.08$ | $0.91 \pm 0.01$ | $0.04 \pm 0.06$ | $0.95 \pm 0.01$ | $0.00 \pm 0.01$ |
| RecGNN | $\mathcal{O}(nm)$ | $1.00 \pm 0.00$ | $1.00 \pm 0.00$ | $0.99 \pm 0.02$ | $0.93 \pm 0.15$ | $0.99 \pm 0.01$ | $0.79 \pm 0.37$ |
| Flood and Echo *random* | $\mathcal{O}(m)$ | $1.00 \pm 0.00$ | $1.00 \pm 0.00$ | $0.97 \pm 0.04$ | $0.77 \pm 0.30$ | $0.98 \pm 0.02$ | $0.48 \pm 0.38$ |
| Flood and Echo *fixed* | $\mathcal{O}(m)$ | $1.00 \pm 0.00$ | $1.00 \pm 0.00$ | $1.00 \pm 0.00$ | $0.99 \pm 0.02$ | $1.00 \pm 0.00$ | $0.89 \pm 0.13$ |

# F  ALGORITHMIC DATASETS

For all the below tasks we use train set, validation set and test set sizes of 1024, 100 and 1000 respectively. The sizes of the respective graphs in the train, validation and test sets are 10, 20 and 100. Performance on this test set demonstrates the model's ability to extrapolate to larger graph sizes. Note that many of the tasks only require the output modulo 2. We reduce the problem to this specific setting so that all numbers involved in the computation stay within the same range, as

otherwise the values have to be interpreted almost in a symbolic way which is very challenging for the learning based models.

**PrefixSum Task** (Grötschla et al., 2022) Each graph in this dataset is a path graph where each node has a random binary label with one marked vertex at one end which indicates the starting point. The objective of this task is to predict whether the PrefixSum from the marked node to the node in consideration is divisible by 2.

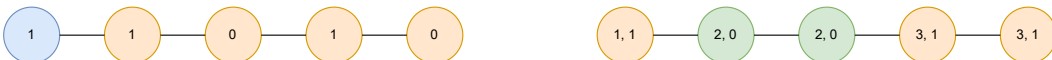

Figure 6: Example graph from the PrefixSum task. The left graph represents the input graph with a binary value associated with each node and the blue node being the starting node. The right graph represents the ground truth solution, each node contains two values the cumulative sum and the desired result which is the cumulative sum modulo 2.

**Distance Task** (Grötschla et al., 2022) In this task every graph is a random graph of $n$ nodes with a source node being distinctly marked. The objective of this task is to predict for each node whether its distance to the source node is divisible by 2.

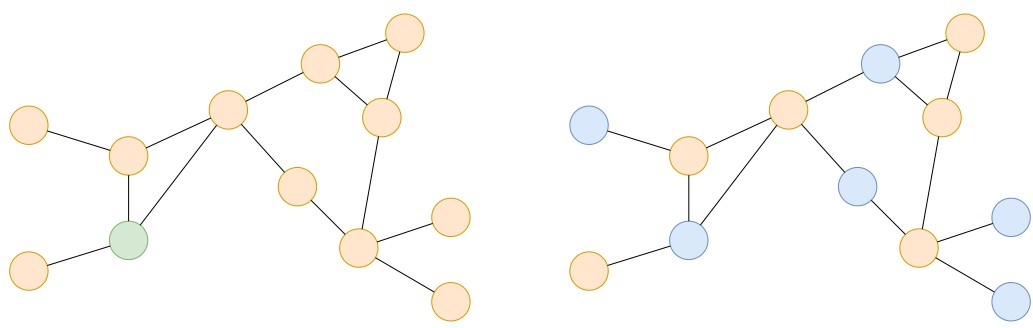

Figure 7: Example graph from the distance task. The green node in the left graph (input graph) represents the source node and the remaining nodes are unmarked. On the right graph (ground truth) all orange nodes are at an odd distance away from the source while the blue nodes are at an even distance away from the source.

**Path Finding Task** (Grötschla et al., 2022) In this task the dataset consists of random trees of $n$ nodes with two distinct vertices being marked separately. The objective of this task is to predict for each node whether it belongs to the shortest path between the 2 marked nodes.

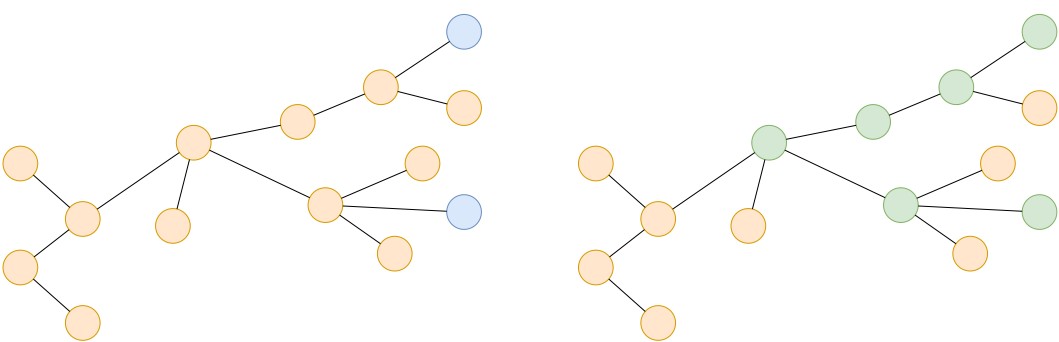

Figure 8: Example graph from the path finding task. The left graph represents the input graph, where the blue nodes are the marked nodes. The right is the corresponding solution where the path between the marked nodes is highlighted in green.

## G  EXPRESSIVE DATASETS

### G.1  DATASETS

**Skip Circles**   (Chen et al., 2023) This dataset consists of CSL(Circular Skip Link) graphs denoted by $G_{n,k}$ which is a graph of size $n$, numbered 0 to $n-1$, where there exists an edge between node $i$ and node $j$ iff $|i-j| \equiv 1$ or $k$ (mod $n$). $G_{n,k}$ and $G_{n',k'}$ are only isomorphic when $n = n'$ and $k \equiv \pm k'$ (mod $n$). Here the number of graphs in train, validation and test are all 10. We can see an example of this construction in Figure 9.

We follow the setup of Chen et al. (2023) where we fix $n = 41$ and set $k \in \{2, 3, 4, 5, 6, 9, 11, 12, 13, 16\}$. Each $G_{n,k}$ forms a seperate isomorphism class and the aim of the classification task is to classify the graph into its isomorphism class by the skip cycle length. Since 1-WL is unable to classify these graphs, we can see in table 1 that the GIN model cannot get an accuracy better than random guessing (10%).

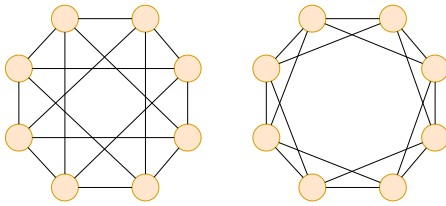

Figure 9: Example graphs from the Skip Circles dataset, namely $G_{n,5}$ and $G_{n,2}$ on the left and the right respectively.

**Limits1 and Limits2**   (Garg et al., 2020) This dataset consists of two graphs from Garg et al. (2020) that despite having different girth, circumference, diameter and total number of cycles cannot be distinguished by 1-WL. For each example the aim is to distinguish among the disjoint graphs on the left versus the larger component on the right. The specific constructions can be seen in Figure 10.

**4-Cycles**   (Loukas, 2020) This dataset introduced by Loukas (2020) originates from a construction by Korhonen & Rybicki (2017) in which two players Alice and Bob each start with a complete bipartite graph of $p = \sqrt{q}$ nodes which are numbered from 1 to $2p$ and a hidden binary key with size being $|p^2|$. The nodes from each graph with the same numbers are connected together. Each player then uses their respective binary keys to remove edges, each bipartite edge corresponding to a zero bit is removed and remaining edges are substituted by a path of length $k/2 - 1$, we use $k = 4$. The

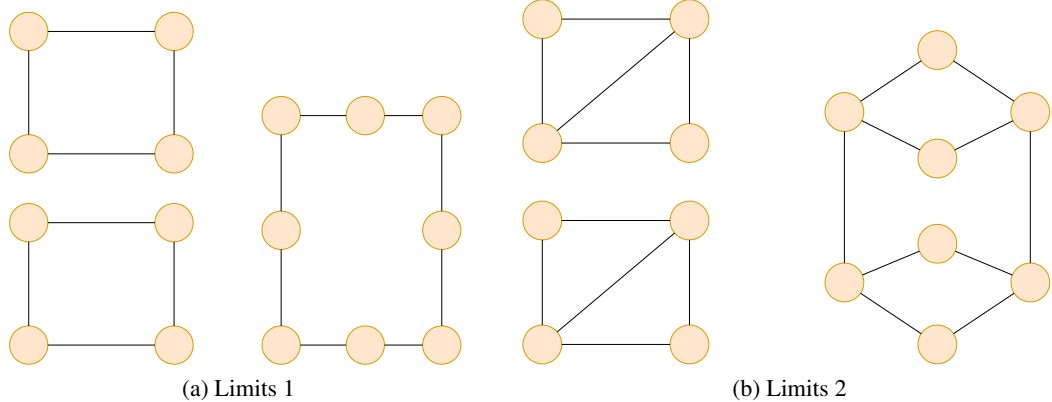

(a) Limits 1                    (b) Limits 2

Figure 10: Counter-examples which MPNNs cannot distinguish from Garg et al. (2020), they cannot distinguish among the graphs in each example.

task is to determine if the resulting graph has a cycle of length $k$. In our implementation the number of train, validation and test graphs we consider are all 25. For a depiction of the construction refer to Figure 11.

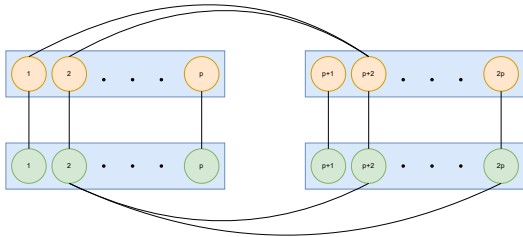

Figure 11: Example construction of Loukas (2020)

, where k=4.

**LLC**  (Sato et al., 2021) This dataset is comprised of random 3-regular graphs and the task is to determine for each node its local clustering coefficient (Watts & Strogatz, 1998) which informally is the number of triangles the vertex is part of. The training and test set are both comprised of a 1000 graphs. The graphs in the train set have 20 nodes, while the graphs in the test set have a 100 nodes testing extrapolation. An example graph from this dataset can be seen in Figure 12.

**Triangles**  (Sato et al., 2021) This dataset akin to the previous contains random 3-regular graphs with the same train/test split and graph sizes. The task here is to classify each node as being part of a triangle or not. An example graph from this dataset can be seen in Figure 12.

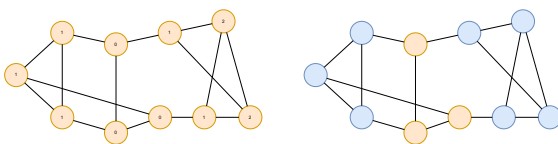

Figure 12: The graphs represent an instance from LLC and Triangles dataset respectively. For the LLC graph(left), each label denotes the ground truth for the graph while for the Triangles(right) graph, the blue nodes are ones which are a part of a triangle, while the orange nodes are not part of any triangle.

