# OpenReview forum: "Flood and Echo: Algorithmic Alignment of GNNs with Distributed Computing"
_ICLR.cc/2024/Conference — Submitted to ICLR 2024_

### Official Review · Reviewer_hXqD · 2023-10-14

**Soundness:** 1 poor
**Presentation:** 1 poor
**Contribution:** 1 poor
**Rating:** 5
**Confidence:** 4

**Summary:**

This paper studies an information propagation scheme, named flood and echo.

In standard GNN message passing, node only exchange information with their immediate 1-hop neighbor in each round. The authors argue that this type of message passing is inefficient with complexity of O(D m ), where D is the diameter of graph and m is the number of edges.

As an alternative, the authors propose “Flood and Echo” which propagates messages in a wave like pattern throughout the entire graph. Starting from a center node, “Flood and Echo”  floods the messages outwards, then the flow reverses and is echoed back. The authors claims this could reduce the complexity to O(m), where m is the number of edges.

The presentation of this paper can be improved. Besides, I have doubt on the complexity part, authors need better explaination on this.

**Strengths:**

This paper aims at proposing a new message passing schema to replace the ordinary GNN's message passing (i.e., each node propagate to its 1-hop neighbors).

The algorithm is inspired by distributed learning. Making connection between different field is interesting.

The authors also explore from 1-WL test based expressiveness.

**Weaknesses:**

It is not clear to me how the computation graph of a single node is constructed in this case. In ordinary GNN, the computation graph is a tree structure, first layer is 1-hop neighbors, second layer is 2-hop neighbors, etc. But I cannot tell how the representation of a single node is computed. If we want the node representation of a node that is not the chosen start node, what is its node representation's computaiton graph?

The flood and echo mainly focused on the forward propagation part. But how its gradient are computed? To compute the gradient for weight parameters, we have to use the “gradient with respect to hidden representations output” and “the input node representation that multiplied to the weight parameters”. In flood and echo, it seems like we have to save the hidden embeddings for each node to compute gradient?

Compared to ordinary message passing GNN, the major difference if the selection of neighbors and the number of propagation steps (aka the number of layers in GNN). In order to propagate information to the full graph, the proposed “flood and echo” requires twice the propagation steps of ordinary message passing GNN due to echo back. I am not sure I understand this correctly and how this can benefit in terms of efficiency.

The "flood" part seems to me is the forward propagation of ordinary GNN, where this GNN's depth is the diameter of the graph?

**Questions:**

- What is FloodConv and FloodCrossConv on line 7 & 8 in Algorithm 1?
- What is phases stands for on line 5 in Algorithm 1?
- The final node representation might be different if the flood start node is chosen randomly each time?
- In the paragraph above Section 4, the authors said “in every run, we only keep the node embedding for the chosen start node”, but why? If we need to compute the embeddings for all nodes in the graph and compute its gradient, shouldn’t we save this for all nodes?

---

> ### Author Response · Authors · 2023-11-15
> **Response 1/2**
>
> We thank the reviewer for his review and address the raised concerns in the following comments.
>
> > It is not clear to me how the computation graph of a single node is constructed in this case. In ordinary GNN, the computation graph is a tree structure, first layer is 1-hop neighbors, second layer is 2-hop neighbors, etc. But I cannot tell how the representation of a single node is computed. If we want the node representation of a node that is not the chosen start node, what is its node representation's computaiton graph?
>
> Indeed, the computation graph of a single node is not straightforward to define as it is in a regular MPNN. In the flooding phase, the update of a single node is influenced by all nodes that lie on the direct path between the node and the starting node. Similarly, in the echo phase, the node embedding is influenced by all nodes for which the node in question lies on the direct path to the root. We have added two more figures in Appendix A which should visualise this and highlight the differences between the two mechanisms.
>
> > The flood and echo mainly focused on the forward propagation part. But how its gradient are computed? To compute the gradient for weight parameters, we have to use the “gradient with respect to hidden representations output” and “the input node representation that multiplied to the weight parameters”. In flood and echo, it seems like we have to save the hidden embeddings for each node to compute gradient?
>
> Note that the main difference lies in the node activations order, but the graph convolutions and computations still stay the same in principle. The main difference is that they happen more sparsely and in a fashion that follows the flooding and echo wave pattern. In practice, the mechanism is implemented in pyg which then makes use of automatic differentiation to get the gradients. The embeddings which need to be stored to compute the gradients are the same as for the regular MPNNs.
>
> > Compared to ordinary message passing GNN, the major difference if the selection of neighbors and the number of propagation steps (aka the number of layers in GNN). In order to propagate information to the full graph, the proposed “flood and echo” requires twice the propagation steps of ordinary message passing GNN due to echo back. I am not sure I understand this correctly and how this can benefit in terms of efficiency.
>
> So if information should be propagated throughout the entire graph and back, then also a regular GNN would need the same number of propagation steps. The Flood and Echo Net only activates a subset of nodes at each propagation step, so even though the number of propagation steps (in terms of layers) is the same, the number of node updates is significantly less (each node is activated a constant number of times instead of O(D) times) and so is the number of messages required to achieve this.
> > The "flood" part seems to me is the forward propagation of ordinary GNN, where this GNN's depth is the diameter of the graph?
> Both the flood and echo are always part of the computation - we have added a new figure in the Appendix A which shows how nodes are activated in the forward pass for both the flood and echo part. A GNN which executes diameter many layers would have a similar reach but with a completely different computation tree (with more updates to every node). We hope this clarification was already helpful, we are not entirely certain what causes the confusion.
>
> > What is FloodConv and FloodCrossConv on line 7 & 8 in Algorithm 1?
>
> In the flood and in the echo parts graph convolutions are applied (the FloodConv), in the implementation, we actually differentiate between edges that connect nodes at a different distance from the starting point (edge from a node at distance d to a node at distance d+1) and nodes at the same distance. Edges which connect nodes that remain at the same distance execute the crossconv convolutions.
>
> > What is phases stands for on line 5 in Algorithm 1?
>
> One execution of a flood followed by an echo is seen as one phase. The Flood and Echo Net could execute multiple such phases.

---

> > ### Author Response · Authors · 2023-11-15
> > **Response 2/2**
> >
> > > The final node representation might be different if the flood start node is chosen randomly each time?
> >
> > The flood and echo mechanism is always the same, but there are different modes, one of which we present here is the random start. Depending on the different starting nodes the individual node representations might indeed differ.
> >
> > > In the paragraph above Section 4, the authors said “in every run, we only keep the node embedding for the chosen start node”, but why? If we need to compute the embeddings for all nodes in the graph and compute its gradient, shouldn’t we save this for all nodes?
> >
> > Again, the mechanism stays the same but in the “all” mode our goal is to only derive the embedding of the node that is marked as the starting node. To save computational effort, we could instead apply the mode with a fixed start where we keep all node embeddings after an execution. One could even think of a mix between the two strategies or come up with more modes in future work.
> >
> > If any open questions remain, please do not hesitate to contact us again. We hope we could properly address and incorporate your suggestions. If this is the case, we would be happy if you could consider raising the score.

---

### Official Review · Reviewer_gPuo · 2023-10-28

**Soundness:** 2 fair
**Presentation:** 2 fair
**Contribution:** 2 fair
**Rating:** 3
**Confidence:** 4

**Summary:**

The work introduces Flood and Echo Net, which is inspired by the design of certain algorithms coming from distributed systems. The work analyzes the procedure from the point of view of expressiveness and efficiency. In regards to efficiency, the authors are particularly interested in how many messages are required to be exchanged. The work provides an empirical evaluation for certain algorithmic benchmarks.

**Strengths:**

The paper proposes a unique message-passing scheme that is inspired by existing literature from distributed systems. The principle of aligning message passing with existing algorithmic paradigms is interesting and was shown by the authors to be fruitful on certain tasks. I also appreciated the theoretical analysis relating the procedure to WL and GIN as it is not obvious that the Flood and Echo network can simulate GIN.

**Weaknesses:**

While the work is interesting I believe there are some weaknesses that need to be addressed.

W1) While the overall design of the Flood and Echo net is motivated by being aligned to distributed systems, it is not motivated why this specific message-passing design may be useful/desired for real-world tasks that likely do not follow such a dynamical aggregation pattern. In other words, the inductive bias given by such a message-passing procedure may be put into question.

W2) While the authors aim to answer "i) How can we enable nodes to gather the required information in a given graph (information exchange), even if is far away", the authors do not relate this to existing work on over-squashing [1, 2]. I believe that while this is not necessary, it would greatly strengthen such claims as the phenomenon is highly related to this question.

W3) The authors evaluate on highly specific benchmarks which seem to be extremely aligned to the Flood and Echo net. It would be more valuable if the authors evaluated on existing algorithmic benchmarks from the field of Neural Algorithmic Reasoning [3].

[1] Understanding over-squashing and bottlenecks on graphs via curvature. Jake Topping*, Francesco Di Giovanni*, Benjamin Chamberlain, Xiaowen Dong, Michael Bronstein. ICLR 2022.

[2] On Over-Squashing in Message Passing Neural Networks: The Impact of Width, Depth, and Topology. Francesco Di Giovanni, Lorenzo Giusti, Federico Barbero, Giulia Luise, Pietro Lio, Michael Bronstein. ICML 2023.

[3] The CLRS Algorithmic Reasoning Benchmark. Petar Veličković, Adrià Puigdomènech Badia, David Budden, Razvan Pascanu, Andrea Banino, Misha Dashevskiy, Raia Hadsell, Charles Blundell. ICML 2022.

**Questions:**

Regarding W1.

Q1) Would the authors be able to give further motivation for the specific design of the message-passing scheme, beyond specific solutions to certain algorithmic problems?

Regarding W2.

Q2) Would the authors be able to provide some comments regarding the over-squashing effect of their model?

Regarding W3.

Q3) Would the authors be able to provide a reason for why they did not evaluate on existing algorithmic benchmarks such as CLRS-30?

Q4) Is the technique designed specifically with only algorithmic tasks in mind or do the authors envision the technique to work on non-algorithmic tasks?

To address W3, it would be important to provide further experimental results on a broader range of tasks.

---

> ### Author Response · Authors · 2023-11-15
>
> We thank the reviewer for his review and address the raised concerns in the following comments.
>
> > Regarding W1. Q1) Would the authors be able to give further motivation for the specific design of the message-passing scheme, beyond specific solutions to certain algorithmic problems?
>
> The main inspiration for the architecture is the flooding and echo design pattern often used in the design of distributed algorithms. On a more conceptual level we believe this pattern to be important as it aligns well with searching (flood) for specific information (in the whole graph when the location is unknown) and then gathering (echo) the specific piece of information back to where the search originated. With the main advantage being that the location does not have to be known in advance or be close as the whole graph can be explored. The motivation is that this searching and gathering pattern would be more generally applicable for settings where nodes are reliant on information exchange which could be done in such a query-like fashion.
>
> > Regarding W2. Q2) Would the authors be able to provide some comments regarding the over-squashing effect of their model?
>
> Thank you for pointing out this issue - We have included such a discussion on over-squashing in our Appendix. We believe that our objective for information exchange is slightly different from the mentioned over-squashing phenomena. In our work, we try to analyse if information can be facilitated through the graph even if it is far away - where the focus lies on the distance that has to be overcome. Here we implicitly assume that there is no topological bottleneck and the main issue for standard GNNs would be that they have to execute many rounds to even distribute the information in question. We are aware that distance and depth are factors that are important for over-squashing. However, as we understand it, over-squashing mainly becomes an issue if there are (topological) bottlenecks in the graph. In this case, the Flood and Echo Net would also struggle and one probably would have to consider some sort of graph rewiring to alleviate the issue - on this modified (rewired) graph, one could then deploy the Flood and Echo Net.
>
> > Regarding W3. Q3) Would the authors be able to provide a reason for why they did not evaluate on existing algorithmic benchmarks such as CLRS-30?
>
> Thank you for pointing out the CLRS-30 benchmark. We would like to test and compare our architecture on the NAR problems that they study in future work. However, currently we see some issues on why we can not apply the Flood and Echo Net. The main obstacle we see is the graph representation of the problems in CLRS. They utilise fully connected graphs as base for their computation, which one side defeats the purpose of Flood and Echo which tries to scale the number of messages sub quadratically. On the other hand, in a fully connected graph, the flood and echo would perform a single activation of all nodes and then echo back. The strengths of the Flood and Echo Net are to exchange information to far away nodes and the mechanism still efficiently generalises to larger graphs - this just does not really make sense if the graph is fully connected as nodes are directly connected and if the graph grows quadratically for larger sizes it cannot be efficient anyway. Lastly, we do hope that we can incorporate some comparisons in the future, as certainly the graph problems which are part of CLRS could be adjusted to make appropriate comparisons. However, as of now there are both conceptual and technical hurdles to overcome in order to perform comparisons in a straightforward way.
>
> > Q4) Is the technique designed specifically with only algorithmic tasks in mind or do the authors envision the technique to work on non-algorithmic tasks?
>
> For now we have focused our analysis of the Flood and Echo Net on its ability to be applied in the setting of extrapolation and information exchange. While algorithmic problems might be the most interesting and easily available setting to study these two properties, we could see future work for applying Flood and Echo Net for the benefit of information exchange in other settings.  For now, we have focused on the presented problems in our work because they allow rigorous control which allows us to gain theoretical insights on the mechanism of the Flood and Echo Net, i.e. the comparison against the theoretical best performance in Section 5.
>
> If any open questions remain, please do not hesitate to contact us again. We hope we could properly address and incorporate your suggestions. If this is the case, we would be happy if you could consider raising the score.

---

> > ### Comment · Reviewer_gPuo · 2023-11-17
> >
> > I would like to thank the authors for the reply.  As it stands the work is conceptually interesting, but is lacking especially in terms of experiments. Unfortunately in my opinion the theoretical aspects of the work do not compensate for this.
> >
> > Regarding the response for Q1, it would be important in my opinion to give concrete examples that are not "algorithmic" in nature in which such a process could be useful. If not, I would recommend focusing on an algorithmic motivation, which could be an interesting motivation regardless.
> >
> > If the latter is the case and this seems to be more likely given the current experimental section, it would be important to find a way to compare with the CLRS-30 benchmarks. I agree that the graphs being fully connected may make this challenging, but it would be at least important to find some tasks in which this would be viable.
> >
> > Unfortunately, I would like to hold my score.

---

> > > ### Author Response · Authors · 2023-11-23
> > >
> > > We thank the reviewer for his comments and we try to address and clarify the points in question.
> > > > Regarding the response for Q1, it would be important in my opinion to give concrete examples that are not "algorithmic" in nature in which such a process could be useful. If not, I would recommend focusing on an algorithmic motivation, which could be an interesting motivation regardless.
> > >
> > > The main aspect of this paper is to introduce the Flood and Echo Net which has a new execution paradigm and should get benefits from its alignment. We believe that this mechanism is interesting because it relates to many aspects that are of great interest in the GNN community such as expressivity or information exchange. Moreover, we think that this mechanism should be very helpful to achieve extrapolation - for which you most probably need to inject the right priors in order to algorithmically align well.
> > >
> > > Therefore, we believe these “algorithmic” tasks and motivation are already the focus of our work as discussed in both the Information exchange and Extrapolation section. We agree that it would be interesting to compare to the CLRS benchmark as they also study extrapolation, however, as outlined in our last response, there are many obstacles to overcome and it is not straightforward to incorporate this and we mainly see this as an interesting way forward.
> > >
> > > To briefly recap, the main focus of our work is to present the novel execution mechanism of Flood and Echo Net. We assess its capabilities and try to fit the new mechanism into the right context in terms of its expressivity, information exchange, and especially its ability to extrapolate. For these aspects, we provide empirical evidence and theoretical analysis.

---

### Official Review · Reviewer_VsT5 · 2023-11-01

**Soundness:** 2 fair
**Presentation:** 3 good
**Contribution:** 1 poor
**Rating:** 3
**Confidence:** 4

**Summary:**

This paper proposes a simple algorithm that mimics the "flood and echo" algorithm in distributed computing as a new computing framework on graphs. The basic idea also is inside part of rooted subgraph GNNs that consider a subgraph with a root node and distance-to-root features. The authors propose this "new" execution framework for GNNs to improve its scalability and its extrapolation ability. The author test its extrapolation ability on some basic tasks comparing with baselines like GIN and RecGNN.

**Strengths:**

1. The authors present the paper in a very easy-to-follow manner. The central idea is simple and easy to get.
2. The angle of viewing propagation mechanisms as execution framework is kind of new, and it seems that the proposed execution framework is valuable in terms of improving extrapolation on mimic some graph algorithms.
3. The author discuss many perspectives of the proposed execution framework like expressivity and information message complexity.

**Weaknesses:**

1. The underlying idea is not new in many perspectives. First, it is studied in rooted subgraph based GNNs, along with distance-to-root feature usage. One can refer to Bohang Zhang's ICML 23 paper. Second, it shares certain similarity to "Agent-based Graph Neural Network". In principle, I feel the author mainly provide another angle of viewing it as kind of execution mechanism of GNNs.

2. The expressivity study/proof is technique-wise simple. Also the author need to discuss the expressivity comparison inside permutation equivalent setting. We don't discuss expressivity for permutation sensitive model as it can achieve universality easily. Hence here the author should focus on Flood And Echo All when discussing expressivity.

3. I feel experimental wise the tasks are limited and datasets are small, baselines are also limited.

**Questions:**

1. The way of "backward and forward" "wave" technique is also used in another paper "A Practical, Progressively-Expressive GNN", where they proved that doing this kind of wave back and forth multiple times towards convergence is the most powerful one. I'm wandering whether the author tested doing the flood and echo multiple times.

2. All this method shares similarity to subgraph GNNs, I'm wandering whether the author can provide some comparison of subgraph GNNs for these extrapolation tasks.

---

> ### Author Response · Authors · 2023-11-15
> **Response 1/2**
>
> We thank the reviewer for his review and address the raised concerns in the following comments.
>
> > The underlying idea is not new in many perspectives. First, it is studied in rooted subgraph based GNNs, along with distance-to-root feature usage. One can refer to Bohang Zhang's ICML 23 paper. Second, it shares certain similarity to "Agent-based Graph Neural Network". In principle, I feel the author mainly provide another angle of viewing it as kind of execution mechanism of GNNs.
>
> We had a closer look at the mentioned paper, which defines Subgraph GNNs the following way: Subgraph GNNs unifiy the view of GNNs which not only rely on their immediate neighbourhood but across different elements, i.e. which one can capture with subgraphs (multi hop neighbourhood or incorporating distance information).
> However, I fail to see how exactly one could express the Flood and Echo Net in this framework. Note that in our work, at each given timestep, only a subset of nodes is currently active and updates their node embeddings. Because these activations happen one after another based on the distance to the starting point, this is similar to a wave pattern. Moreover, because these activations (and node updates) happen one after another, information can propagate through the graph following the pattern. The mechanism of a node A which is closer to the starting point updating itself before a node B that is farther away (the updated A embedding might even have an influence on B) is central to the idea of the Flood and Echo Net.  We have added an additional Figure in the Appendix A to highlight the execution mechanism more clearly.
>
> Currently we do not see any way of expressing the same dynamics in the framework of subgraph GNNs - and therefore, I believe they are not that closely related. However, we have added an extended paragraph about the connection in the Appendix which discusses subgraph GNNs.
>
> We agree that the paper shares some similarity to Agent-based Graph Neural Networks (AgentNet) in the sense that we propose a different computation mechanism compared to more standard GNNs. Which is also a reason why we mention the paper in our related work section. However, there are important key differences between the two proposed methods. In AgentNet there are a fixed number of agents which traverse the graph structure similar to a learned random walk - whereas in Flood and Echo Net, all nodes (not only where an agent is currently active) perform computations determined by the wave activation pattern.
>
> We think thatFlood and Echo Net goes beyond “another angle of viewing” as it defines a very specific computation flow which is inherently different from standard MPNNs and other execution paradigms such as the aforementioned AgentNet.
>
> > The expressivity study/proof is technique-wise simple. Also the author need to discuss the expressivity comparison inside permutation equivalent setting. We don't discuss expressivity for permutation sensitive model as it can achieve universality easily. Hence here the author should focus on Flood And Echo All when discussing expressivity.
>
> We understand the concerns of this particular point. Note, that the execution mechanism for the Flood and Echo Net always stays the same, regardless of the chosen operation mode and our analysis still holds. Moreover, the proposed scheme does not break permutation equivariance, but introduces an additional tie-break in distinguishing the starting node, which is a lot weaker than universality (i.e. by including random features). Finally, note that we provide the expressivity experiments for the Flood and Echo All mode in Table 1. Even in the “all” mode it is more expressive - you can consider the example in Figure 2 where every node (no matter the starting node) can distinguish the two graphs.

---

> > ### Author Response · Authors · 2023-11-15
> > **Response 2/2**
> >
> > > I feel experimental wise the tasks are limited and datasets are small, baselines are also limited.
> >
> > The main reason for choosing the specific experiments throughout the empirical evaluation of our paper was to demonstrate and verify the proposed architecture in terms of its expressiveness, ability to exchange information and extrapolation to larger instances not encountered during training. While the task setting is synthetic in nature, this allows us great control over the setup, where we can exactly analyse and provably argue about the information which needs to be exchanged.
> > For the extrapolation comparison, we for now considered a regular MPNN (GIN) and a recurrent baseline (RecGNN) which can adapt the number of rounds depending on the task. We felt that many additional baselines would not give additional insight, due to their inability to adapt the number of rounds which is required to exchange sufficient information in the extrapolation setting for message passing on the original graph topology. On the other hand, baselines which make use of higher order message passing usually come at a great computational cost, making it hard to scale to larger graphs for extrapolation. If you have any specific baselines in mind that make sense in the described context I would be happy to consider them for comparisons.
> >
> > >  The way of "backward and forward" "wave" technique is also used in another paper "A Practical, Progressively-Expressive GNN", where they proved that doing this kind of wave back and forth multiple times towards convergence is the most powerful one. I'm wandering whether the author tested doing the flood and echo multiple times.
> >
> > Thank you for pointing out this work, we were not aware of it and have included a reference in our own work. Indeed it is possible to execute multiple phases of the Flood and Echo Net. In fact, we execute 2 phases in the experiments presented in the extrapolation section which we also mention in the main text.
> >
> > >  All this method shares similarity to subgraph GNNs, I'm wandering whether the author can provide some comparison of subgraph GNNs for these extrapolation tasks.
> >
> > Note that Flood and Echo Net operate on the original graph topology, which has the advantage that its mechanism is relatively simple and computationally inexpensive (compared to higher order message passing). Comparing subgraph GNNs which would consider a higher order neighbourhood might not be a really fair comparison. Moreover, the main reason why we did not incorporate this comparison is that the extrapolation experiments provably require the exchange of information over a distance of O(D). So if a subgraph GNN only considers its O(1) neighbourhood, I don’t think it will be able to learn anything sensible in these tasks. As a comparison we have included other baselines such as RecGNN which is aware of this issue and scales the number of rounds to overcome this limitation. If you have any concrete suggestion on a subgraph GNN which would still make sense in this context I am happy to have another look  to see if we can include them in our comparisons.
> >
> > If any open questions remain, please do not hesitate to contact us again. We hope we could properly address and incorporate your suggestions. If this is the case, we would be happy if you could consider raising the score.

---

> > > ### Comment · Reviewer_VsT5 · 2023-11-22
> > > **Response to author's rebuttal**
> > >
> > > I want to thank the author for the detailed response and further figures provided in appendix for presenting the designed algorithm. I want to explain my standpoint more clear. I acknowledge that the execution mechanism, if looking at specifically, is not existing. However the underlying reason that it improves expressivity and performance, is the same as rooted subgraph. For a giving starting point, the reason why the designed algorithm work is because of these identifiability provided by a single rooted subgraph. The author may need to understand the theory behind the rooted subgraph based GNN to see where the expressivity power comes from. I really appreciate the authors additional effort, however I don't think my evaluation is incorrect. I would like to keep the score.

---

> > > > ### Author Response · Authors · 2023-11-23
> > > >
> > > > We thank the reviewer for his comments and we try to address and clarify the points in question.
> > > > > I acknowledge that the execution mechanism, if looking at specifically, is not existing.
> > > >
> > > > If we understand it correctly, the reviewer acknowledges that the proposed execution mechanism is novel (=is not existing?). If so, thank you - we believe that this different execution mechanism is the main point of the paper. The analysis that follows tries to adequately assess this mechanism in the context of expressivity, information exchange and extrapolation.
> > > >
> > > > > However the underlying reason that it improves expressivity and performance is the same as rooted subgraph. For a given starting point, the reason why the designed algorithm works is because of the identifiability provided by a single rooted subgraph.
> > > >
> > > > Regarding the reason why the proposed mechanism improves expressivity, we agree with the reviewer. The information provided by the starting node which breaks some symmetry in the graph is the main driver to go beyond the 1WL limitation. One can also see this as a marking policy or connect it to rooted subgraphs, although keep in mind that in our case Flood and Echo always considers the entire graph rather than a subgraph and it is inherently part of the execution.
> > > >
> > > > So the underlying reason is similar, but how this provided symmetry breaking translates to actual performance and increased expressivity in a GNN is also very much dependent on the execution mechanism. The Flood and Echo Net’s main aim is to align the computation - and our analysis is meant to put the Flood and Echo Net in the right context regarding expressivity as it is a newly proposed mechanism. Therefore, we believe that the main insight from the expressivity section should be that this new aligned execution provides and can make use of a symmetry breaking which then is more expressive than 1WL.
> > > >
> > > > Regarding the performance through the other experiments - we do not believe that it stems from increased expressivity. For example, in the information exchange prefix sum task, one end of the graph is marked (so it also provides a marked node for all other baselines) but it is still very challenging to exchange the information or achieve extrapolation. Here, our proposed Net can overcome this hurdle and we can even analyse how close its performance compares to the theoretical upper bound.

---

### Official Review · Reviewer_W7g1 · 2023-11-02

**Soundness:** 4 excellent
**Presentation:** 4 excellent
**Contribution:** 4 excellent
**Rating:** 8
**Confidence:** 3

**Summary:**

This paper applies flood and echo, a distributed computing algorithm, to graph-based learning tasks. Theoretical and empirical analysis verifies that the proposed method enjoys better expressiveness than massage-passing NN.

**Strengths:**

- This paper presents a novel technique for graph-based learning tasks.
- Experiments show that flood and echo algorithm achieves overwhelming advantages.
- A novel task (PrefixSum task) is introduced in this paper for verifying the expressiveness of the proposed method.
- Theoretical analysis is provided.

**Weaknesses:**

- The implementation of the proposed framework might be complex as it is not supported by the existing graph learning frameworks such as DGL and PyG.
- The computation is more expensive than GNN. Each round of full-graph traverse can only compute one node's embedding.

**Questions:**

- How to implement the proposed algorithm?

- Typo: 'ob' in the first paragraph of section 3

- This is a curious question. The idea of Flood and Echo is borrowed from distributed computing. Can we apply it to distributed GNN training/inference? What are the benefits we can expect?

---

> ### Author Response · Authors · 2023-11-15
>
> We thank the reviewer for his review and address the raised concerns in the following comments.
>
> > The implementation of the proposed framework might be complex as it is not supported by the existing graph learning frameworks such as DGL and PyG.
>
> The implementation of the proposed Flood and Echo Net might seem more complex at first than it actually is. The non-trivial part is to ensure the correct message flow, which can be done using proper masking so that the entire model can still run efficiently on the GPU. We have implemented the architecture in PyG and will release the entire code base upon publication.
>
> > The computation is more expensive than GNN. Each round of full-graph traverse can only compute one node's embedding.
>
> Note that for all presented modes the execution of a Flood and Echo Net always computes embeddings for all nodes. The difference lies in the chosen starting point and what embeddings you want to keep for the final prediction. For some mode, such as the “all” mode, it might make sense to run an execution for every node as the start once and only keep its own embedding after the run. However, we want to stress that just from a computational point of view an entire phase of the Flood and Echo net is as expensive as a single round of a standard MPNN (using O(m) messages).
>
> > This is a curious question. The idea of Flood and Echo is borrowed from distributed computing. Can we apply it to distributed GNN training/inference? What are the benefits we can expect?
>
> The Flood and Echo execution pattern only ever activates a few nodes at a time, maybe this could be leveraged during inference time for more efficiency. If you want to focus on distributed training/inference in the sense of scaling to multiple machines we do not see an immediate connection. In this work, we are mainly inspired by the algorithm design done in distributed computing. For distributed execution on multiple machines we think this concerns a different area of distributed computing.
>
> If any open questions remain, please do not hesitate to contact us again. We hope we could properly address and incorporate your suggestions. If this is the case, we would be happy if you could consider raising the score.

---

### Meta-Review · Area_Chair_wUHZ · 2023-12-05

**Metareview:**

This is a very interesting contribution which improves algorithmic alignment in graph neural networks through the use of a "flood and echo" activation pattern. While I find that the Authors are on the right path, and this work certainly has potential for a top-tier conference, it is also evident that this work could have been better evaluated and presented, and its novelty needs to be better justified. At this point, the work would certainly benefit from another round of revision, and I need to recommend rejection.

**Justification For Why Not Higher Score:**

While there is not a full consensus on the rating for this work, the sole accepting reviewer did not oppose rejection, and indeed recommended the Authors to revise the work for another round of reviews, in light of the other Reviewers' comments and concerns. I am in agreement with this conclusion.

**Justification For Why Not Lower Score:**

N/A

---

### Decision · Program_Chairs · 2024-01-16

Reject